# Green Graphene–Chitosan Sorbent Materials for Mercury Water Remediation

**DOI:** 10.3390/nano10081474

**Published:** 2020-07-28

**Authors:** Ana Bessa, Gil Gonçalves, Bruno Henriques, Eddy M. Domingues, Eduarda Pereira, Paula A. A. P. Marques

**Affiliations:** 1Centro de Tecnologia Mecânica e Automação (TEMA), Mechanical Engineering Department, University of Aveiro, 3810-193 Aveiro, Portugal; arcb@ua.pt (A.B.); ggoncalves@ua.pt (G.G.); eddy@ua.pt (E.M.D.); 2Centro de Estudos do Ambiente e do Mar (CESAM) & Department of Chemistry, University of Aveiro, 3810-193 Aveiro, Portugal; brunogalinho@ua.pt; 3Laboratório Associado para a Química Verde-Rede de Química e Tecnologia (LAQV-REQUIMTE) & Department of Chemistry, University of Aveiro, 3810-193 Aveiro, Portugal; eduper@ua.pt

**Keywords:** removal of Hg, real waters, chitosan, graphene oxide-based nanocomposites

## Abstract

The development of new graphene-based nanocomposites able to provide synergistic effects for the adsorption of toxic heavy metals in realistic conditions (environment) is of higher demand for future applications. This work explores the preparation of a green nanocomposite based on the self-assembly of graphene oxide (GO) with chitosan (CH) for the remediation of Hg(II) in different water matrices, including ultrapure and natural waters (tap water, river water, and seawater). Starting at a concentration of 50 μg L^–1^, the results showed that GO–CH nanocomposite has an excellent adsorption capacity of Hg (II) using very small doses (10 mg L^–1^) in ultrapure water with a removal percentage (% R) of 97 % R after only two hours of contact time. In the case of tap water, the % R was 81.4% after four hours of contact time. In the case of river and seawater, the GO–CH nanocomposite showed a limited performance due the high complexity of the water matrices, leading to a residual removal of Hg(II). The obtained removal of Hg(II) at equilibrium in river and seawater for GO–CH was 13% R and 7% R, respectively. Our studies conducted with different mimicked sea waters revealed that the removal of mercury is not affected by the presence of NO_3_^–^ and Na^+^ (>90% R of Hg(II)); however, in the presence of Cl^–^, the mercury removal was virtually nonexistent (1% R of Hg(II)), most likely because of the formation of very stable chloro-complexes of Hg(II) with less affinity towards GO–CH.

## 1. Introduction

Water pollution is one of the most severe problems our planet faces, and managing water resources sustainably under climate change, population growth, and economic development poses novel challenges in the 21st century, termed the Century of the Environment [1,2]. Heavy metals are an extremely hazardous class of nonbiodegradable and bioaccumulative elements that present a serious threat to all forms of life, even at trace levels concentrations [3]. Among the heavy metals, the United States (US) Government Agency for Toxic Substances and Disease Registry ranks mercury (Hg) as the third most toxic element to human health [4]. Aquatic ecosystems are an important part of the global biogeochemical cycle of Hg. Inorganic Hg can convert to toxic methyl Hg form (CH_3_Hg) in water (river and sea), driving one of the main human exposure path by the consumption of fish, particularly if caught in the sea [5]. Once it reaches the food chain, Hg tends to bioaccumulate, ultimately causing harmful effects to human health [6,7]. Therefore, Hg presents a significant risk to both the global environment and human health alike, as it is well expressed in Europe Comission’s environment report [8].

In this context, there is a high demand to develop advanced green materials and technologies with high performance for water remediation. Nanotechnology has earned wider attention and the use of engineered nanomaterials show promise for global water treatment [9,10]. One can find many examples in the literature of natural-based materials used as sorbents for several prejudicial metal(loid)s, such as a myriad of tree leaves, agricultural biowaste, or even banana peels [11,12,13].

Recently, new graphene-based three-dimensional (3D) macrostructures with enhanced sorption capability towards hazardous organic molecules and heavy metal(loid)s have been proposed [14,15,16,17,18]. Graphene consists in a single atomic layer of graphite with bond length between carbon–carbon of 0.142 nm and, like graphite, the atoms in graphene are arranged in a hexagonal lattice. As there are only sp^2^ carbon atoms on the graphene sheets, pristine graphene can only provide van der Waals force to bind adsorbates, so it is not a good adsorbent for many types of contaminations, such as metal ions. Graphene oxide (GO) sparked huge interest among researchers, as it retains plenty of the properties of the highly valued pristine graphene, but it is much easier and cheaper to make in bulk quantities and easier to process. It is composed of planar graphene-like aromatic domains (sp^2^) of random size separated by sp^3^-hybridized carbons decorated by hydroxyl, epoxy, and carboxyl groups [19]. GO is amphiphilic and has a negative charge, and its surface is comprised of multiple oxygen-containing groups. This multitude of oxygen functional groups allows an incredible variety of chemical interactions with other molecules. For example, there are hydrogen bonds between hydroxyl groups from GO and other hydroxyl-rich molecules; electrostatic interactions between the negatively ionized carboxyl groups located at the edges and positive charged species; π–π interactions with other π-conjugated materials due to the delocalized electrons over sp^2^-hybridized carbon atom domains [20]. The chemical composition and molecular structure make GO behave like a polymer [21]. Therefore, the self-assembly of 2D GO nanosheets in the presence of natural polymers into 3D macrostructures is possible due to the establishment of previously described chemical and/or physical bonds and can result in the formation of an hydrogel [17]. This is a relatively simple method which can be easily scaled up for commercial or industrial use [18]. Recently, our group developed a 3D GO structure by chemically modifying the GO surface with nitrogen functional groups (3DGON) increasing its removal efficiency towards Hg up to 95% [22].

A sensible polymer selection for the development of the above-mentioned GO-based 3D macrostructures would focus on the presence of amine functional groups due to the high affinity of these groups and the oxygen functional groups from GO. Additionally, these polymers show a natural high sorption ability for different pollutants [23]; thus, this combination may result in a synergistic behavior for improved adsorption of several pollutants in water. A widely accessible natural biopolymer obtained from the alkaline deacetylation of chitin, chitosan (CH), is hydrophilic, environmentally friendly, and nontoxic and has been used as an agent for the removal of dyes and heavy metals from contaminated waters [24]. For the specific case of heavy metal(loid)s, the adsorption onto GO–CH-based materials has been studied for the removal of Cr [25,26,27,28,29,30], Cu [25,31,32,33,34], Pb [25,31,32,35,36,37], As [32,38], Au, and Pd [39] or U [28]. GO–CH was also applied to remove Hg [40], although only in deionized water and for relatively high concentrations of material (1 g L^–1^) and Hg (up to 500 mg L^–1^), which in addition to not being environmentally friendly, are not representative of real cases.

An easy and environmentally friendly way to obtain GO–CH would be by the formation of hydrogels by self-assembly, which was already reported by recent works in the literature [31,33,41]. In this case, CH acts as a crosslinking agent, and the presence of amine groups in the positively charged CH strongly attract the negatively charged GO through electrostatic interactions. Additionally, hydrogen bonds should also occur between both GO and CH, creating stable hydrogels, which can be converted into aerogels by freeze drying. Nowadays, most of the reports in the literature are restricted to explore GO-CH nanocomposite materials for heavy metals adsorption in batch studies with ultrapure water. Evaluation of these graphene-based sorbents under real wastewater conditions is thus needed to provide critical information regarding their real applicability on water remediation.

The aim of this study was the use of a simple, eco-friendly, and easily scalable preparation of a GO–CH nanocomposite and its applicability in Hg removal from different natural water matrices, starting at more realistic relatively low concentration of Hg (50 μg L^–1^), which corresponds to the maximum allowable limit in wastewater discharges from industry in Europe [42]. GO–CH nanocomposites showed an excellent adsorption performance and efficiency for Hg in ultrapure water and tap water (*q*_e_ = 4554 and 4700 μg g^–1^, respectively). However, in other natural waters, where the complexity of the matrices is much higher, a significant decrease on the adsorption ability of the nanocomposite material was observed, particularly seawater assays.

## 2. Materials and Methods

### 2.1. Materials Synthesis

Chitosan (CH) with M.W. 310000–375000 (Sigma-Aldrich, Saint-Louis, MO, USA), solution with a concentration of 5 mg mL^–1^ was prepared in distilled water and acetic acid (1% *v/v*). Graphene oxide (GO) water dispersion (0.4 wt% concentration from Graphenea, San Sebastian, Spain) was added to the polymer solution with a targeted ratio of 24% *v/v* (GO–CH). This ratio was optimized in order to achieve the most stable hydrogel. The pH of both GO and CH solutions were modified to 2 before mixing, using 0.1 mol dm^–3^ NaOH (Sigma-Aldrich) or HCl (Sigma-Aldrich) solutions. After the mixture, the resulting solution was quickly shaken for 10 s to form the hydrogel, which was subsequently freeze-dried (Telstar LyoQuest HT-40, Beijer Electronics Products AB, Malmoe, Sweden) at −80 °C, obtaining 3D porous structures. The lyophilized samples were then washed in MQ water (ultrapure water from Milli-Q^®^, Burlington, MA, USA; 18 MΩ cm^–1^) for 12 h to remove acidic residues. Finally, the washed sample was freeze-dried once more and stored until the adsorption experiments.

### 2.2. Materials Characterization

XPS spectra were obtained in an ultra-high vacuum (UHV from SPECS, Berlin, Germany) system with a base pressure of 2 × 10^–10^ mbar. The high-resolution spectra were taken at normal emission take-off angle and with a pass energy of 20 eV, which provides a global instrumental peak broadening of about 0.5 eV. The spectra were rectified in binding energy by referencing to the first component of the C1s core level at 284.5 eV (Csp^2^). The chemical structure of the GO–CH aerogels, GO and CH, were analyzed via attenuated total reflectance Fourier transform infrared (ATR-FTIR) in a Bruker Tensor 27 FT-IR spectrometer (Bruker Corporation, Billerica, MA, USA). The spectra were obtained between 4000 and 400 cm^–1^, using a resolution of 4 cm^–1^ and 256 scans. The microstructure of the GO–CH scaffolds was observed using scanning electron microscope (SEM) in a Hitachi SU-70 (Tokyo, Japan) operating at 15 kV. The zeta potential was measured, at different pH, in a Malvern Panalytical Nano-ZS Zetasizer (Malvern, UK). The specific surface area of the GO–CH aerogel was determined by means of N_2_ sorption isotherms, using the Brunauer–Emmett–Teller (BET) method, in a Gemini Micromeritics device (Norcross, GA, USA). The results of the BET study are presented in the Appendix A.

### 2.3. Water Collection and Characterization

For this work, different types of waters (MQ, tap, river, and sea) were used to evaluate the efficiency of Hg removal by GO–CH in natural conditions. The Vouga river was the source for the river water (40°40′42″ N, 08°22′18″ W), whereas the seawater was collected at the beach in Vagueira, near Aveiro (40°32′58″ N 8°46′31″ W). Tap water was sourced at the University of Aveiro. The full physicochemical characterization of the waters used has already been reported elsewhere [43,44]. After being filtered (0.45 μm pore size filter), all the waters used in this work were left for at least 24 h to pre-equilibrate after spiking with Hg before joining the sorbent. The pH of the contaminated solutions was ~4.0, 7.1, 4.3, and 7.8 for MQ, tap, river, and sea waters, respectively. The elemental composition of the major and minor elements of the untapped matrix was determined by inductively coupled plasma (ICP, Jobin–Yvon JY70 Plus Spectrometer, Edison, NJ, USA).

### 2.4. Hg Sorption Studies

The study of the sorption of Hg onto GO–CH was executed in lot experiments, where ~10 mg of CO–CH was added to 1 L of Hg spiked water for up to 24 h contact in Schott glass bottles under consistent magnetic stirring (700 rpm). A certified standard Hg solution (1001 ± 2 mg L^−1^ of Hg(II) in HNO_3_ 0.5 mol L^−1^, from Merck (Darmstadt, Germany) was used to spike the ultra-pure (MQ), tap, river, and sea water with a concentration of 50 µg L^–1^, which matches the maximum value allowed for Hg in discharges (Directive 84/156/EEC) [42]. It should be noted that European Union imposed that Hg emissions, discharges and losses to water cease or be phased out by 2021 (Directive 2013/39/EU 2013) [45]. The sorption kinetics was evaluated by removing 5 to 10 mL of water at pre-determined periods of time, followed by centrifugation at 5000 rpm for 3 min. The supernatant liquid was then transferred to a small Schott glass bottle (25 mL) and acidified to pH ≤ 2 using Suprapur^©^ HNO_3_ (65% *v/v*) from Merck (Darmstadt, Germany). The amount of Hg was quantified using cold vapor atomic fluorescence spectroscopy (CV-AFS), in a PSA 10.025 Millennium Merlin Hg analyzer (Orpington, UK) with SnCl_2_ (2% *m/v* in HCl 10% *v/v*) as a reducing agent. All tests were made in duplicate and a control experiment, consisting in Hg contaminated water without GO–CH, was always running simultaneously.

#### 2.4.1. Analysis of Sorption Data

The performance of a removal process (% R) was evaluated by calculating the percentage of sorbate (Hg) removed from the solution by the sorbent (GO–CH) using the following mathematical expression:(1)$$R\left(\%\right)=\frac{C_{0}-C_{t}}{C_{0}}\times 100$$
where *C*_0_ represents the original Hg concentration in solution (μg L^–1^) and *C*_t_ is the Hg concentration at a given time *t* (μg L^–1^). Presuming that all the removed Hg remained in the sorbent, one can estimate the concentration of Hg in the material at time *t*, *q_t_* (μg g^–1^) using:(2)$$q_{t}=\frac{\left(C_{0}-C_{t}\right)}{m}\times V$$
where *V* (L) is the volume of the solution and *m* (g) the mass of sorbent. When the equilibrium is reached, Equation (2) can be rewritten taking into account *t* = *t_e_*, *q_t_* = *q_e_* and *C_t_* = *C_e_* [22]. In most cases, the actual initial concentration of the spiked solutions show small deviations from the nominal initial concentration; thus, in order to compare results between trials, the analysis of the results was presented in terms of standard concentrations *C_t_*/*C_0_*.

#### 2.4.2. Kinetics and Equilibrium Models

In this study, the kinetics of Hg uptake by GO–CH was followed using three reaction models in their nonlinear form to fit the experimental data [46]—specifically, Lagergren pseudo-first-order model [47], Ho’s pseudo-second-order model [48], and the Elovich model [49] (Appendix A). For a more detailed study, two diffusion-based models were also applied, namely, Boyd’s film-diffusion [50] and Weber’s pore-diffusion [51] models (details in Appendix A).

## 3. Results and Discussion

### 3.1. Chemical and Structural Analysis

The synthesis of the nanostructured composite GO–CH was completed by simply shaking the mixture of the GO and CH solutions. The strong electrostatic attraction among the two components of the mixture lead to the formation of stable gels at room temperature. After gelation, GO–CH was lyophilized to obtain the respective foams in dry state or aerogel. Figure 1 depicts SEM images of GO–CH nanocomposite at different magnifications, detailing the typical structural macroporosity of these aerogels, which has an alveolar-like microstructure.

Figure 2 compares the FTIR spectra for GO, CH, and GO–CH, showing only the details below 2000 cm^–1^, for greater clarity. The GO exhibits the typical hydroxyl peak at 3440 cm^–1^ (O–H stretch), 1402 cm^–1^ (CO–H symmetry stretch), carboxyl peaks at 1722 cm^–1^ (C=O stretch) and 1037 cm^–1^ (C–O stretch), and the peak at 1607 cm^–1^, which can be attributed to the epoxy group (COOH asymmetry stretch) and also to sp^2^ carbon skeletal network (C=C stretch) [52,53]. The spectra for CH exhibits a broad peak centered at ~3257 cm^–1^ (–NH_2_ extension, –OH stretching, and inter-hydrogen bonds); a peak at 2868 cm^–1^ (–CH stretch); a peak at 1631 cm^–1^, which corresponds to the NHCO group (C=O stretch); and a peak at 1543 cm^–1^, which can be attributed to NH_2_ bending mode. The peak at 1405 cm^–1^ can be assigned to the –CH bend, while the one at 1150 cm^–1^ can be assigned to C–O–C asymmetric stretching. The bands at 1062 cm^–1^ and 1026 cm^–1^ can be attributed to skeletal vibrations of C–O (stretching), while the peak at 897 cm^–1^ corresponds to the pyranoid ring stretching mode [26,52,54]. In the case of the GO–CH nanocomposites, it is possible to observe that the band corresponding to C–O stretching mode is relatively more intense than for the GO due to the presence of –OH groups from chitosan. The lower relative intensity of the peak at 1717 cm^–1^ (C=O stretching) than for GO confirms the interaction, at least partially, of the carboxylic groups of GO with the amine group of chitosan [52].

### 3.2. Mercury Sorption Studies

The reduction of Hg concentration as a function of time was followed in four distinct water matrixes (MQ, tap, river, and sea water) in the presence or absence (control) of GO–CH, as depicted in Figure 3. The concentrations of Hg in the control solutions were observed to be relatively constant over the whole test period, which indicates that a possible loss of Hg due to the adsorption in the glass walls or due to volatilization were insignificant. Under the presence of GO–CH sorbents, there was an evident decrease in the concentrations of Hg, *C_t_/C_0_*, from 1.0 to <0.1 in MQ water, and from 1.0 to 0.2 in tap water (Figure 3). The results suggest a strong interaction between the active sites at the surface of the nanocomposite and the Hg ions. At the beginning of the experiments, the surface of the sorbent material is free of metal, thus inducing a high concentration gradient between the solution and the nanocomposite. In MQ water, after the initial rapid descent, the sorption equilibrium is reached at *t* = 2 h. In tap water, Hg sorption rate is slower, reaching the equilibrium at *t* = 4 h, as the driving force decreases. In the case of river and sea waters (Figure 3), GO–CH was not efficient in reducing the Hg contamination. This observation emphasizes the importance of studying metal adsorption in real waters, which is overlooked in most of the studies, leading to potentially misleading affirmations as for the real potential of the studied materials.

Figure 4 depicts the efficiency of GO–CH in the Hg decontamination process, in terms of maximum removal percentage (% R) and the time needed to reach equilibrium, which was contrasted to the legal values for the presence of Hg in water [45,55,56]. In MQ and tap water, the sorption kinetics showed to be very fast, which is promising in view of a practical application [57].

At the equilibrium, after only 2 h, the remaining Hg in the liquid phase for the system with GO–CH in MQ water was 1.3 μg L^–1^ (in line with the European guideline for drinking water quality of 1 μg L^−1^) [45]. Kysas et al. also used a GO–CH composite to remove Hg from MQ water and showed similar efficiency (~95% R) and equilibrium time, but starting from a much higher metal concentration (100 mg L^–1^), which corresponds to a much higher residual Hg concentration after the equilibrium was reached (~5 mg L^–1^). In the case of tap water, the residual Hg concentration was 9.5 μg L^−1^_,_ but this value was, nevertheless, attained after only 4 h of contact time. This value is closer to that of the World Health Organization (WHO), of 6 μg L^−1^ for inorganic mercury for the case of a 60-kg adult consuming 2 L of water a day and distributing 10% of the total daily intake (TDI) to drinking water [58].

However, for river and sea water, GO–CH revealed to be inefficient in removing Hg from such matrices, with concentrations at equilibrium of 43.5 and 46.5 μg L^−1^, respectively.

### 3.3. Kinetic Modelling

Represented in Figure 5 are the experimental values of Hg sorbed onto GO–CH as μg of Hg adsorbed per g of GO–CH over a period of time (24 h) or *q_t_* (μg g^–1^), as well as the adjustments made by the kinetic models of pseudo-first-order (PFO), pseudo-second-order (PSO), and Elovich.

The quality of the fits and the values of the kinetic parameters resulting from the modelling are shown in Table 1. In general, for MQ and tap water, the models used were deemed suitable to describe the kinetics of Hg sorption on GO–CH (0.9379 < *R*^2^ < 0.996). In the case of river and sea water, the low removal values and the high variability in the results led to poor fitting parameters (R^2^ < 0.8289). For GO–CH in MQ water, the model that fits the most is the PFO model, which can be confirmed by the higher values of *R*^2^ (0.996) and low Sy.x values (139). Nevertheless, this model slightly underestimated the value of *q_e_* (PFO = 4544 and exp. = 4554).

On the contrary, the kinetic model with the worst performance in the adjustments was the Elovich model (with values of *R*^2^ = 0.9533 and Sy.x = 328). When PFO represents the experimental points better, the surface of the sorbent is deemed to be homogeneous, and in theory, only one binding mechanism is possible [59]. Therefore, it is proposed that, in the process of Hg adsorption onto GO–CH in MQ water, there is no restriction of surface area and binding sites, and thus, the adsorption process resembles an homogeneous one. In the case of tap water, the model that most closely fits the data is PSO, with *R*^2^ = 0.99 and Sy.x = 161, but PFO (*R*^2^ = 0.9789 and Sy.x = 234) shows a closer *q_e_* value to the experimental values than PSO (PFO = 4623, PSO = 4899, and exp. = 4700).

To gather more information on the Hg sorption mechanism onto GO–CH, namely, on most probable rate-controlling steps, a piecewise linear regression (PLR) [60] of the adsorption data was performed based on the film-diffusion model proposed by Boyd [50] and the intraparticle-diffusion model proposed by Weber [51]. In all the water matrices, Boyd’s method showed a first linear segment that included the origin, which is a sign that film-diffusion is not the rate-defining step in this sorption process [61,62]. In the case of Weber’s model, the Akaike Information Criteria (AIC) were used to compare the fits (to determine the most probable number of segments), indicating a two-step fitting as the more likelihood, for all the water matrices. Therefore, the postulation that the sorption takes place in two diffusion steps is the most likely to be correct. Figure 6 shows the experimental (exp.) points (full symbols) overlapped by the estimated (est.) values (lines) for all the water matrices and the kinetics parameters are presented in Table 2. The first linear segment corresponds to a step where the diffusion is faster due to higher concentration gradient, which can correspond to the diffusion in the larger pores. The second segment can be ascribed to the equilibrium stage with a much less marked slope, which is an indication that the intra-particle diffusion is reduced as the concentration gradient decreases.

### 3.4. Removal Mechanism

To study the sorbents surface charge, the zeta potential of GO–CH in MQ water was measured in different pH (2–12) and the results are shown on Figure 7. GO–CH zeta potential shows that it has an amphoteric nature, with a point zero charge (PZC) of 6.7, where the CH primary amines are protonated for lower pH. This trend is in accordance to previous reports for similar materials [30,34,63].

Speciation studies in ultrapure water (pH = 4) suggested that Hg in solution is mainly present in the neutral form: 93.0% Hg(OH)_2_, and has a diminutive contribution of positive ions 5.5% Hg(OH)^+^ and 1.5% Hg^2+^ (determined by Visual MINTEQ 3.1). Interestingly, for GO–CH in MQ (pH 4.0) or tap water (pH 7.1), the removal efficiency is quite high, even if the surface is mostly protonated, as shown by the zeta potential study (Figure 7). Thus, it is anticipated that, for these waters, and accordingly to the kinetic study, the removal mechanism of these GO-based macrostructures is mainly governed by physical sorption, where intermolecular attractions, such as induced dipole–dipole interactions, should be predominant. In the case of river water (pH 4.3), the removal efficiency drops, which can be linked to the presence of different ions (see Appendix A), and may compete directly with Hg for the sorbents active sites, together with the existence of some organic matter which can interact with the Hg ions. At the pH of sea water (pH 7.8), GO–CH does not adsorb much Hg, despite showing a negative zeta potential, although the formation of Hg chloro-complexes must be considered in this process. With this respect, further tests were conducted for interpretation of the experimental data observed for seawater solutions (see Section 3.5).

The XPS survey spectra of the GO–CH nanocomposites are shown in Figure 8, before (black line) and after (red line) the contact with Hg solution, in MQ water. Figure 8a shows the anticipated peaks for O1s, C1s, and N1s, attributed to the structural composition of the two components in the nanocomposite materials. A residual S2s peak can also be detected, most probably a remaining from the synthesis of GO. While the oxygen and carbon peaks result from the contribution of both GO and CH, naturally, the nitrogen peaks are attributed only to the structural contribution of the CH. The high-resolution XPS spectra (HR-XPS) with curve fitting for C1s of GO–CH nanocomposites (Figure 8b) shows the presence of the C=C groups at 284.5 eV and C–O/C–N functional groups at 286.4 eV. The HR-XPS spectra with curve fitting for N1s (Figure 8c) display peaks at 398.8 eV that correspond to the presence of the free amine groups of the polymer and at 401.3 eV that correspond to N^+^ species, suggesting the formation of weak electrostatic interactions between the primary amines of the polymers and oxygen functionals groups of GO [64,65]. After the exposure of GO–CH to the Hg solution, the strong interactions of the amine groups with Hg can be witnessed in the HR-XPS for N1s (Figure 8c), where one can observe the reduction in the relative intensity of the peak at 401.3 eV with a simultaneous increase of the peak at 398.8 eV [66], and also a slight positive shift of the peaks by 0.5 eV [67]. These indications imply that the sorption process occurs primarily by a charge transfer from N to Hg, making N groups the main sorption sites and that the efficiency of the sorption is controlled by its protonation [68]. According to the hard and soft acids and bases (HSAB) theory, the softer acid mercury ions should have strong affinity to softer bases like amines, resulting in the formation of metal-chelate complexes [69].

The XPS survey of the composite materials (Figure 8a) after the contact with Hg clearly show the presence of the characteristic Hg peak at ~100 eV. The HR-XPS (Figure 8d) shows peaks at 100.9 eV (Hg4f_7/2_) and 104.7 eV (Hg4f_5/2_), which can be attributed to the oxidized state of Hg [22,70], confirmed by the predominance of Hg(OH)_2_ and Hg(OH)^+^ at pH 4.0 (Figure 7).

C1s and O1s peaks show much higher relative intensity (Figure 8a), which represents the majority of the components of GO–CH and no significant changes before and after the contact with Hg solution was detected. Nevertheless, the influence of these elements on the whole sorption process should not be ignored. In fact, in a previous publication by this group, it was shown that the oxygen functional groups have an important contribution to the adsorption mechanism of Hg species in MQ matrix [22].

### 3.5. Influence of Coexisting Ions in Natural Waters

ICP was used to characterize the elemental composition of the natural waters spiked with Hg before and after contact with GO–CH (Appendix A). The major elements detected in the waters (Ca, Na, K, and Mg) presented similar values before and after Hg sorption experiments. In terms of minor elements, in tap water, the most expressive reduction due to the presence of GO–CH occurred for Al, Fe, and Zn, whereas in river water, only Fe and Pb concentrations were significantly reduced. In sea water, only Al was reduced to half its initial concentration and the remaining elements remained virtually the same.

The removal of Hg by GO–CH in river and sea water was strongly inhibited, which cannot be explained through ζ potential. The pH of the sea water spiked with Hg was 7.8, above the PZC of GO–CH of 6.7, turning the surface of the macrostructure negatively charged, which in theory should favor the removal of Hg cations, unfortunately not observed. In sea water, there was a higher presence of cations, principally, Na, Mg, K, and Ca, as well as a higher concentration of Cl^–^, which may hinder the ability of GO–CH to adsorb Hg due to an increased competition for the binding sites at the surface of the macrostructure, or mainly due to changes in the mobility and speciation of Hg (formation of negatively charged chloro-complexes). To clarify this point, experiments conducted in 0.5 M NaCl and 0.5 M NaNO_3_ (Figure 9), mimicking the ionic strength of sea water, showed that the removal of Hg by GO–CH is practically unaffected by the presence of NO_3_^–^ and Na^+^ (93% R of Hg in 0.5 M NaNO_3_ vs. 98% R of Hg in MQ water), the latest being the major cation in sea water. On the other hand, in the presence of Cl^–^, the removal was virtually nonexistent (1% R of Hg), most likely because the formation of very stable chloro-complexes of Hg with less affinity towards GO–CH.

## 4. Conclusions

This study focuses on a simple and eco-friendly synthesis of a GO–CH nanocomposite to obtain an aerogel (after lyophilization) and used it to remove Hg in ultrapure (MQ) and natural waters (tap, river, and sea water). Starting at a concentration of Hg 50 μg L^–1^, which matches to the maximum permissible wastewaters discharges values for Hg, 10 mg L^–1^ of the material was able to reduce the level of this contaminant down to 1.3 μg L^–1^ in MQ, and 9.5 μg L^–1^ in tap water. The equilibrium was achieved after 2 h for MQ water and after 4 h for tap water. However, for river and sea water, the GO–CH nanocomposite was not efficient in removing Hg. It was confirmed that the higher concentration of Cl^–^ in aqueous medium can lead to the formation of chloro-complexes with reduced affinity to GO–CH. Nevertheless, GO–CH shows a good potential as sorbent for mercury in MQ and tap water, and the insights revealed in this study for the remaining water matrices should be considered for future rational design of nanocomposite materials for natural water remediation.

## Figures and Tables

**Figure 1 nanomaterials-10-01474-f001:**
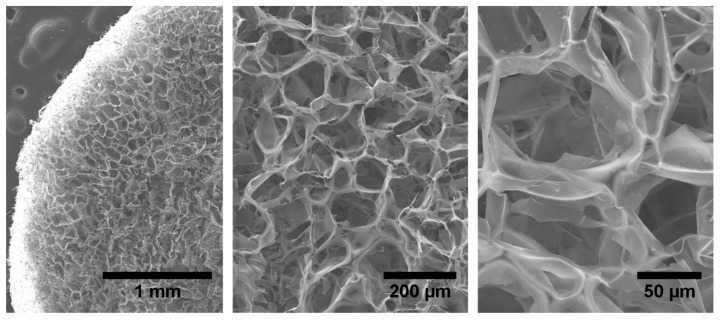
Details of SEM micrographs at different magnifications of the prepared GO–CH aerogels.

**Figure 2 nanomaterials-10-01474-f002:**
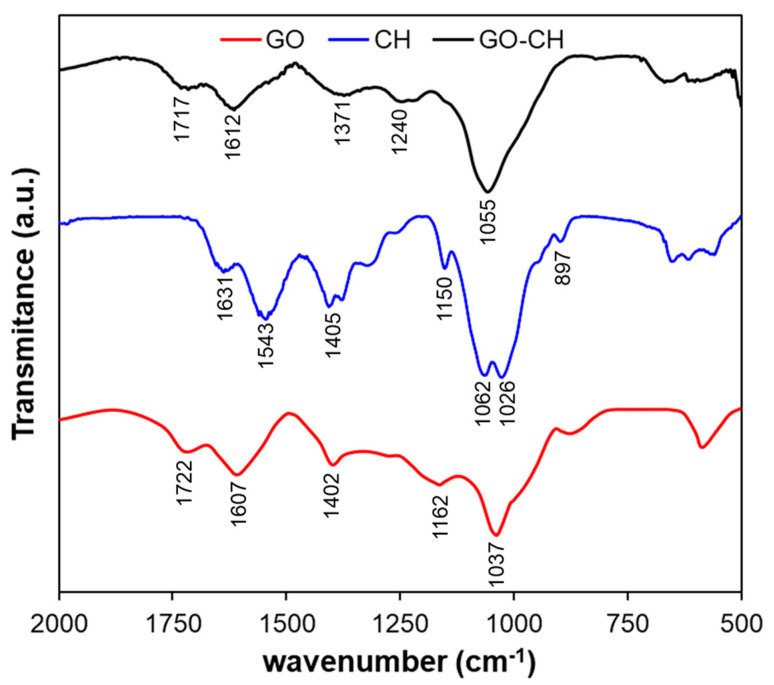
FTIR spectra for GO, CH, and the GO–CH nanocomposite.

**Figure 3 nanomaterials-10-01474-f003:**
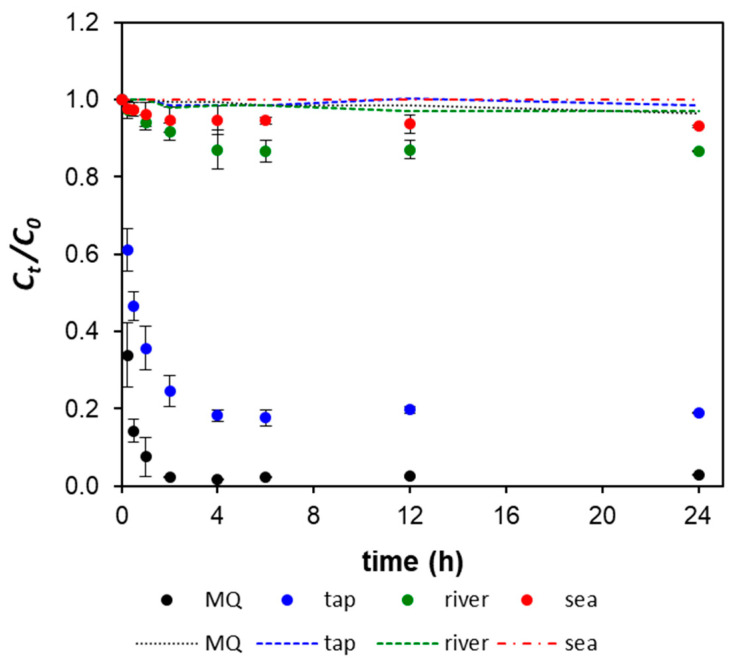
Evolution of the normalized concentration of Hg in solution (*C_t_/C_0_*) as a function of time (h) for ~10 mg L^–1^ of GO–CH nanocomposite in different water matrixes contaminated with Hg. The dotted lines represent the control Hg spiked waters. Initial concentration of Hg where ~50 µg L^–1^. The results correspond to mean ± standard deviation of two replicates.

**Figure 4 nanomaterials-10-01474-f004:**
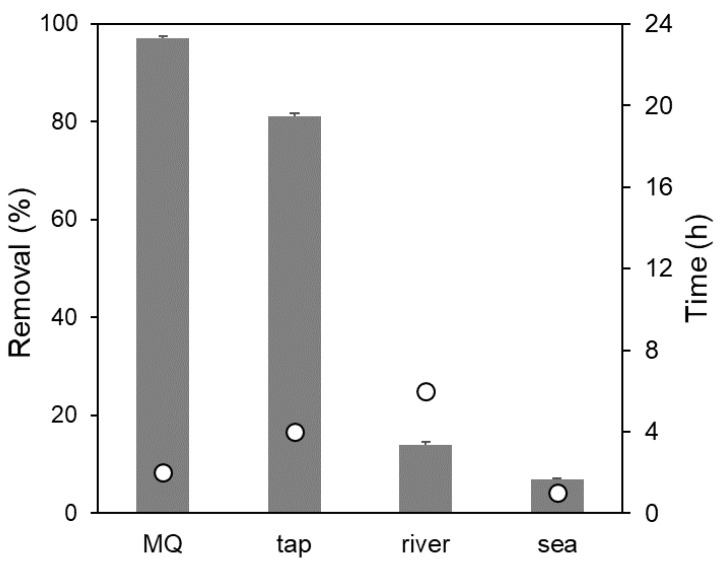
Efficiency of GO–CH in the removal of Hg from contaminated MQ, tap, river, and sea waters: (bars) percentage of Hg removed at equilibrium and (circles) time for equilibrium. The original concentration of Hg was 50 µg L^–1^; the amount of GO–CH used was 10 mg L^–1^. The results are the mean values of two replicates.

**Figure 5 nanomaterials-10-01474-f005:**
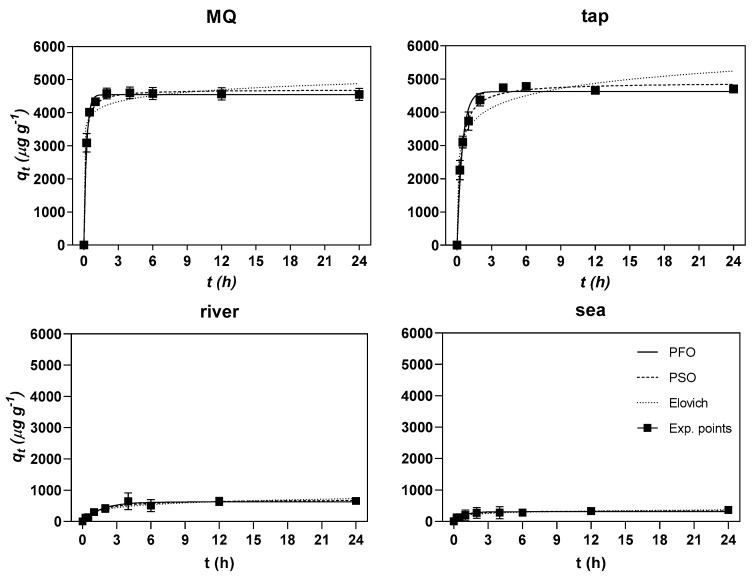
Experimental points of the PFO, PSO, and Elovich models to experimental data regarding the sorption of Hg onto GO–CH over time. The concentration of GO–CH and Hg(II) in the matrices were of ~10 mg L^−1^ and ~50 µg L^–1^, respectively.

**Figure 6 nanomaterials-10-01474-f006:**
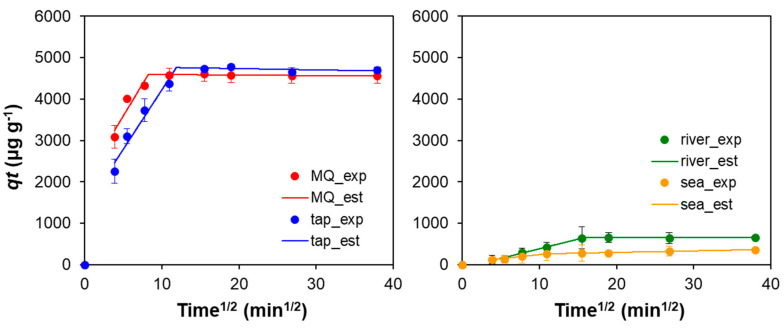
Best-fit results for the kinetic modeling of Hg sorption onto GO–CH using Weber’s intraparticle-diffusion model for all the matrices used in this work. Full symbols represent the experimental data, while the segmented lines represent the estimated values.

**Figure 7 nanomaterials-10-01474-f007:**
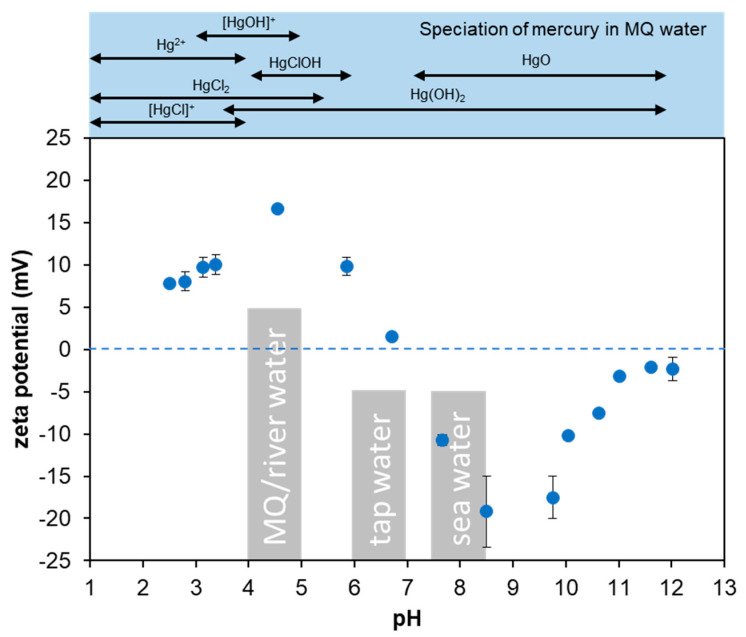
Zeta potential for the GO–CH aerogels at different pH. The grey boxes represent the pH of each type of water at which the experiment was conducted. At the top, one can find the speciation for Hg in MQ water as a function of pH.

**Figure 8 nanomaterials-10-01474-f008:**
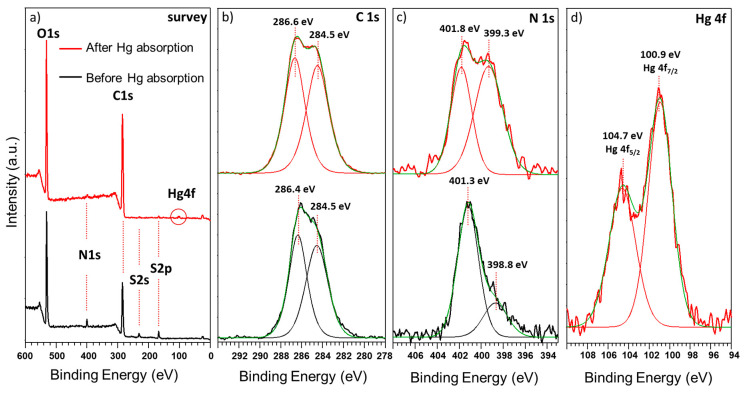
(**a**) XPS survey spectra of GO–CH before and after contact with Hg(II) solution (MQ water), HR-XPS of the (**b**) C1s, (**c**) N1s, and (**d**) Hg 4f peaks and respective deconvolutions.

**Figure 9 nanomaterials-10-01474-f009:**
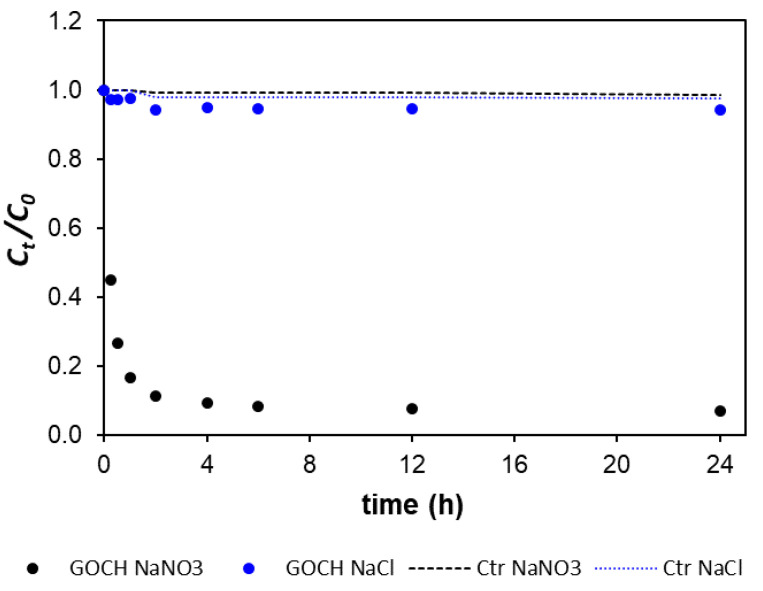
Variation of the normalized concentration of Hg in solution (*C_t_/C_0_*) as a function of contact time (h) with ~10 mg L^–1^ of GO–CH in 0.5 M NaCl and 0.5 M NaNO_3_ solutions. Controls correspond to the absence of sorbent. The original concentration of Hg(II) was ~50 µg L^–1^.

**Table 1 nanomaterials-10-01474-t001:** Fitting parameters for the several reaction models applied to the adsorption of Hg on GO–CH for the water matrices in this study. Experimental *q_e_* were also added for comparison.

Models	Water Matrices			
	MQ	Tap	River	Sea
*q_e1_* exp. ± SD (μg g^–1^)	4554 ± 180	4700 ± 66	656 ± 16	360 ± 19
**Pseudo First Order**				
*q_e1_* ± SD (μg g^–1^)	4544 ± 40	4623 ± 74	627 ± 47	310 ± 32
*k_1_* ± SD (h^–1^)	4.44 ± 0.24	2.20 ± 0.16	0.61 ± 0.17	1.19 ± 0.49
*R^2^*	0.9960	0.9789	0.8289	0.5973
Sy.x	139	234	113	91.1
**Pseudo Second Order**				
*q_e2_* ± SD (μg g^–1^)	4698 ± 58	4899 ± 6 3	711 ± 67	342 ± 41
*k_2_* ± SD (h^–1^)	0.0019 ± 0.0002	0.0007 ± 0.00005	0.001 ± 0.0004	0.0045 ± 0.003
*R^2^*	0.9877	0.9900	0.8256	0.6217
Sy.x	168	161	114	88.3
**Elovich**				
ß ± SD (g μg^–1^)	0.0038 ± 0.0008	0.0019 ± 0.0002	0.0071 ± 0.002	0.019 ± 0.006
α ± SD (μg g^–1^ h^–1^)	1.36 × 10^9^ ± 4.49 × 10^9^	413,852 ± 360,766	1069 ± 640	2046 ± 2505
*R^2^*	0.9533	0.9379	0.7928	0.6279
Sy.x	328	401	124	87.6

**Table 2 nanomaterials-10-01474-t002:** Kinetic parameters resulting from the use of Weber’s intraparticle-diffusion model to fit the experimental sorption data of Hg onto GO–CH.

Matrix	Stage	Breakpoint (min)	*K*_i_ (μg g^–1^ h^–1/2^)	*R* ^2^
**MQ** **2**	1	68	307.7	0.8647 (*n* = 4)
	2	-	–1.285	0.6257 (*n* = 5)
**tap** **2**	1	141	286.7	0.9534 (*n* = 5)
	2	-	–3.062	0.3554 (*n* = 4)
**river** **2**	1	247	46.45	0.9854 (*n* = 5)
	2	-	0.1609	0.0467 (*n* = 4)
**sea**	1	120	20.53	0.8743 (*n* = 4)
	2	-	3.663	0.9536 (*n* = 5)

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
