# Peer review of "Green Graphene–Chitosan Sorbent Materials for Mercury Water Remediation"

_nanomaterials, 2020, doi:10.3390/nano10081474_

Round 1

Reviewer 1 Report

The manuscript is presenting very interesting topic of the formation of green graphene-chitosan sorbents for mercury remediation.

The removal of heavy metals and particularly the toxic ones are essential for the environment. The mercury is absolutely the example of such metal.

The introduction section is clear and logical. However, the last part, where normally the goals are presented there is something like a small summary. The authors should clearly define the aims.

Section 2.1. – please add the country of the origin of chemicals, equipment etc.

In the manuscript, there are small editing errors that should be corrected. For instance, line 28 problem, not problems; line 94 oC not need to underlie; line 337 NO3- 3 need to be in subscript

Were these materials used as scaffolds for other applications (line 106)?

The details about the applied techniques should be added. SEM – did authors use any sputtering?

And please pay attention to the information presented in the data. There are data for ICP; however, there are no details about the machine, technique, sample preparation.

Page 4 – really, all the bands were assigned to the symmetric vibrations?

Fig. 4 and Fig. 6 – do not connect the experimental points. Either show the trend line of leave not connected. And please add the error bars.

Fig. 5 – it will be easier to see differences if all the y-axis will be in the same range. Please make all of them till 6000.

Fig. 7 – There are no error bars for all experimental points.

What is the stability of these materials? How many times is it possible to use? Please add suitable information.

Reviewer 2 Report

The authors synthesized a "green" oriented carbonaceous composite material in order to study its removal efficiency against Hg in various aqueous samples. The work is well defined and the results derived from the comparison of the different aquatic samples interesting. In the light of a minor revision required prior publication, the authors can elevate their work slightly by the following suggestions:

  1. No evidence exist within the characterizations that graphene exists. It is a bulky material, so the use of graphite is safer.
  2. It should be mentioned at least the targeted ration per mass of the two counterparts and why this strategy of materials design was followed.
  3. The use of graphite/graphite oxide as a filler for the synthesis of composite materials is a well-known approach. It should be pointed out based on literature, which can be the potential beneficial effects. 
  4. Comparison with other "green" derived materials reported as adsorbent of Hg should be included, like: Removal of heavy metals by leaves-derived biosorbents (doi: 10.1007/s10311-018-00829-x), Agricultural biomass/waste as adsorbents for toxic metal decontamination of aqueous solutions (10.1016/j.molliq.2019.111684).
  5. Some suggestions on how to overcome the fact that the presence of Cl can block the adsorption capability should be discussed. 

Reviewer 3 Report

The authors report the development of GO-chitosan composite for water remediation when the Hg (II) concentration is water is very low (50 ppb) in ultrapure, river and sea waters. At those conditions, the selectivity towards Hg(II) should be very high. Therefore, the capacity and equilibrium constants of Hg(II) adsorption values are very important and they are not reported.

As chitosan has amino groups. It will interact with many transition metals. In order to determine the selectivity towards Hg(II) adsorption studies with other metals present in considerable concentration in the water to be treated should be studied.

Fig 1 shows that the GO-CH aerogels have high porosity which is very important for the kinetic of metal adsorption; however, Fig 3 shows very slow kinetics explain why. The values of surface area and pore size distribution is missing, these experiments are important for materials for adsorption studies.

Line 181 what is the meaning of robust interaction?. usually strong and weak terms are used. These s are related to the equilibrium of the adsorption however; no equilibrium studies were performed.

Round 2

Reviewer 1 Report

The manuscript has been corrected 

Reviewer 3 Report

In order to maintain the standard of the journal, the information reported is not enough and a study of the interaction of the material with Hg ions should be improved. Such us adsorption capacity, equilibrium constant of interaction with Hg ions and other ions should be included. The studies at low concentration and in different waters should be a part of the work, and the results explained by the experiments carried out with single metals.

In figure 7, in sea water, the concentration of Cl- is very high, it can form complexes with Hg(II) therefore, the speciation will be different.

Line 185 What is the meaning of “at t =4h,as the driving force decreases “

3.3 Kinetic modelling

Kinetic modelling was reported using data from figure 5. The more important values are those before the equilibrium specially the first minutes where the changes in adsorption are more marked; but, only 4 points in each experiment can be considered meaningful values. In addition, the points have big error. Therefore, with the quality of data reported, the results obtained from the modelling have not meaning.

3.4 Removal mechanism

Line 278 “removal mechanism is governed by physical sorption” considering that Hg(OH)2 is at pH 4. However, it is stated in line 297 that XPS studies suggest the strong interaction of the amine group with Hg.